# Medium-Chain Fatty Acids Rescue Motor Function and Neuromuscular Junction Degeneration in a *Drosophila* Model of Amyotrophic Lateral Sclerosis

**DOI:** 10.3390/cells12172163

**Published:** 2023-08-28

**Authors:** Ella Dunn, Joern R. Steinert, Aelfwin Stone, Virender Sahota, Robin S. B. Williams, Stuart Snowden, Hrvoje Augustin

**Affiliations:** 1Centre for Biomedical Sciences, Department of Biological Sciences, Royal Holloway University of London, Egham, Surrey TW20 OEX, UK; ella.dunn.2021@live.rhul.ac.uk (E.D.); robin.williams@rhul.ac.uk (R.S.B.W.); 2Faculty of Medicine & Health Sciences, Queen’s Medical Centre, Nottingham NG7 2UH, UK; joern.steinert@nottingham.ac.uk (J.R.S.); aelfwin.stone@nottingham.ac.uk (A.S.)

**Keywords:** ALS, MCFA, 4-MOA, NA, *Drosophila*, motor function, NMJ, glutamate

## Abstract

Amyotrophic lateral sclerosis (ALS) is an adult-onset neurodegenerative disease characterised by progressive degeneration of the motor neurones. An expanded *GGGGCC (G4C2)* hexanucleotide repeat in *C9orf72* is the most common genetic cause of ALS and frontotemporal dementia (FTD); therefore, the resulting disease is known as C9ALS/FTD. Here, we employ a *Drosophila melanogaster* model of C9ALS/FTD (C9 model) to investigate a role for specific medium-chain fatty acids (MCFAs) in reversing pathogenic outcomes. *Drosophila* larvae overexpressing the ALS-associated dipeptide repeats (DPRs) in the nervous system exhibit reduced motor function and neuromuscular junction (NMJ) defects. We show that two MCFAs, nonanoic acid (NA) and 4-methyloctanoic acid (4-MOA), can ameliorate impaired motor function in C9 larvae and improve NMJ degeneration, although their mechanisms of action are not identical. NA modified postsynaptic glutamate receptor density, whereas 4-MOA restored defects in the presynaptic vesicular release. We also demonstrate the effects of NA and 4-MOA on metabolism in C9 larvae and implicate various metabolic pathways as dysregulated in our ALS model. Our findings pave the way to identifying novel therapeutic targets and potential treatments for ALS.

## 1. Introduction

Amyotrophic lateral sclerosis (ALS) is an adult-onset, neurodegenerative disease affecting around 2 in 100,000 individuals worldwide [1], characterised by a progressive loss of motor neurones in the spinal cord that results in paralysis [2,3]. The survival of ALS patients is largely dependent on disease progression, clinical presentation, respiratory function and nutritional status [4]. The major symptoms of ALS are related to motor dysfunction, such as muscle weakness and spasticity (stiffness), and dysphagia (difficulty swallowing) [3]. A decline in walking ability is a strong predictor of death in ALS patients [5], measured using a 6 min or 10 m walking test [6,7]. It has been noted that interventions that improve gait, such as moderate exercise, can increase survival in ALS mouse models [8] and slow disease progression in ALS patients [9]. Hybrid assistive limb (HAL) treatment, in which a wearable robot supports the patients’ gait, is able to significantly improve the 10 min walk test in ALS patients, indicating that long-term treatment may help patients to maintain their walking speed [10] and potentially improve their survival rate. Therefore, therapies able to improve motor function in ALS may be advantageous for both treating the symptoms and increasing survival in patients. 

Synaptic dysfunction is another prominent feature of ALS, intimately linked to reduced motor function, where synaptic degeneration and loss are commonly seen in the cortex and corticospinal motor neurones in ALS patients [11,12]. In particular, the NMJ, a specialised cholinergic synapse permitting signalling between muscle and nerve, is known to be dysfunctional in ALS [13]. Mouse models of ALS demonstrate that loss of NMJ circuitry is an early hallmark of the disease, preceding motor neurone degeneration [14,15]. Confirming these results in human ALS patients is not always easy, as motoneuronal samples are difficult to obtain. However, NMJ denervation has been identified in muscle biopsies [16] and autopsy samples from early-stage ALS patients [17], indicating that NMJ disruption is an early feature of the disease. Consequently, treatments combatting NMJ dysfunction could be beneficial in preventing or delaying ALS onset or preventing disease progression. It is worth noting that impaired motor phenotypes are often seen alongside NMJ dysfunction, as a reduction in neuromuscular transmission results in skeletal muscle weakness [18]. Therefore, it is possible that any treatments affecting motor function in ALS may also impact the NMJ physiology. 

The medium-chain triglyceride ketogenic diet (MCT-KD) is a high-fat, low-carbohydrate diet, that was initially thought to utilise medium-chain fatty acids (MCFAs) as a primary energy source, although several pharmacological mechanisms have been proposed for specific fats [19]. Although traditionally used to treat drug-resistant epilepsy, the MCT-KD is gaining traction as a treatment for neurodegenerative diseases including Alzheimer’s disease (AD) and Parkinson’s disease (PD) [19,20,21]. There is currently limited evidence to suggest that the MCT-KD may be beneficial in patients with ALS [22]. However, the MCFA octanoic acid (OA) can improve motor symptoms in an ALS mouse model by protecting against motor neurone loss, indicating the potential benefits of this class of molecules in targeting ALS motor symptoms [22,23]. Interestingly, related MCFAs, such as nonanoic acid (NA) and 4-methyloctanoic acid (4-MOA) were initially identified through reproducing the effect of an established epilepsy treatment, valproate, on phosphoinositide signalling [24], and subsequently shown to provide potent anti-seizure activity in both ex vivo rat hippocampal slice experiments and in an in vivo status epilepticus model [25]. Related compounds also directly inhibit AMPA (α-amino-3-hydroxy-5-methyl-4-isoxazolepropionic acid) receptor activity [26], suggesting the potential clinical efficacy of these compounds in the treatment of epilepsy. Since NA and 4-MOA provide much greater efficacy in seizure control than octanoic acid (OA), they may be more effective in ALS treatment. 

This study aims to evaluate the effects of specific MCFAs on locomotor activity, and synaptic strength and morphology, in a *Drosophila* model of ALS. We have chosen to model the most common genetic cause of ALS—a hexanucleotide repeat expansion in *C9orf72* [27], accounting for 40–50% of familial and 5–10% of sporadic ALS cases [28]. This mutation is also the most common genetic cause of frontotemporal dementia (FTD) [27], responsible for 20–25% of familial and 6–10% of sporadic cases [29]; the resulting disease is therefore referred to as C9ALS/FTD [27]. *Drosophila* models of C9ALS/FTD (C9 models) are well characterised; accumulating evidence suggests that arginine-rich (GR) toxic dipeptide repeats (DPRs), produced via the non-ATG (RAN) translation of hexanucleotide *GGGGCC* (G4C2) repeats in both the sense and antisense directions and in all reading frames, drive C9ALS/FTD pathogenicity [30,31,32,33,34,35]. The expression of poly-GR constructs in *Drosophila* motor neurones impairs larval locomotion and induces degeneration at the NMJ [31,35], making it a suitable system for modelling the functional aspects of ALS. 

We show that NA and 4-MOA improve larval locomotion and rescue multiple morphological and physiological defects at the NMJ in a *Drosophila* C9 model. We demonstrate the ability of NA to modify the postsynaptic density of glutamate receptors, and the ability of 4-MOA to restore dysfunctional presynaptic vesicular release at the C9-larval NMJ. Whilst both MCFAs improve motor dysfunction and NMJ defects in the ALS model, the effects of these compounds are not identical. NA improves morphological defects at the NMJ, whereas 4-MOA impacts NMJ physiology, indicating that although they are the same class of molecule, they do not share a mechanism of action. Furthermore, we analysed the effects of NA and 4-MOA on metabolism, demonstrating that both compounds partially rescue the metabolic dysregulation seen in C9 larvae. We hypothesise that NA and 4-MOA act on the GABA glutamate shunt, nicotinamide/nicotinate metabolism, aspartate, alanine and asparagine metabolism, the urea cycle and pyrimidine metabolism in the C9 larvae, implicating these pathways as dysregulated in the ALS metabolism. 

## 2. Materials and Methods

### 2.1. Fly Stocks and Husbandry

All *Drosophila melanogaster* were fed a standard 1 × SYA (Sugar, Yeast, Agar) food mixture consisting of 1.5% agar, 10% sugar, 10% yeast, 3% Nipagin and 0.3% propionic acid, made up to volume with d.H_2_O. All *Drosophila* stocks were kept in vials/bottles of standard 1 × SYA food in incubators at either 18 °C (maintenance) or 25 °C on a 12 h:12 h light:dark cycle at constant (60%) humidity. All flies were reared at a standard larval density, of around 20–40 flies per vial. Eclosing virgin female flies and adult male flies were collected at 2 h intervals during the 12 h “light” period of the light:dark cycle, using a small amount of CO_2_ as an anaesthetic. 

Expression of poly-GR DPRs was achieved with the *GAL4-UAS* system (targeted gene expression via GAL4-dependent upstream activator sequence) [36]. The *D42-GAL4* line (stock #8816) was obtained from the Bloomington Drosophila Stock Centre (BDSC). The *UAS-GR100* line (100 copies of GR dipeptide repeats) was a gift from T. Niccoli at University College London (UCL). The *w1118* line was sourced from L. Partridge at UCL. 

### 2.2. NA and 4-MOA

Nonanoic acid (NA, Alfa Aesar, B21568) (Ward Hill, USA) and 4-methyloctanoic acid (4-MOA, SAFC, STBB5288) were dissolved in DMSO and added to standard 1 × SYA food to make final concentrations of 5 mM and 100 μM, respectively. *Drosophila* virgin females and adult males were left to mate in compound-free food for 48 h before transferring into fresh vials containing compound-enriched food. The 1 × SYA food containing 0.1% DMSO was used as a control. 

### 2.3. Behavioural Phenotypes

Motor function in *Drosophila* larvae was measured using two related behavioural tests: larval locomotion and body contraction frequency assays. Third-instar wandering larvae were selected, washed in a mesh basket using d.H_2_O, and transferred to an 80 mm petri dish containing solidified 1% agarose gel. Larvae were allowed 30 s for acclimatisation before conducting the assays. For larval locomotion (crawling) assays, the petri dish was placed on top of graph paper and the number of 1 mm^2^ squares that the larvae crossed were counted for 1 min. For larval body contraction frequency assays, the number of body wall contractions (defined as a body wall contraction from anterior to posterior) were counted for 1 min. 

### 2.4. Immunocytochemistry and Confocal Microscopy

Third-instar wandering larvae were manually dissected in ice-cooled phosphate buffered saline (PBS [50 mM phosphate buffer, 685 mM NaCl, 13.5 mM KCl]) on a dissection plate (10 parts silicone elastomer base, 1 part silicone elastomer curing agent) and fixed in Bouin’s fixative (7.5 mL saturated picric acid, 2.5 mL formalin, 0.5 mL glacial acetic acid) for 20–30 min. Type IIA glutamate receptors (GluRIIA) were detected using mouse monoclonal anti-GluRIIA primary antibody (DSHB, 8B4D2) at 1:100, and visualised using goat anti-mouse polyclonal AF488-labelled secondary Ab (ThermoFischer, A-110011) (Waltham, MA, USA) at 1:400. Presynaptic neurones were detected and visualised using polyclonal cyanine 3 (Cy3)-labelled anti-horseradish peroxidase (HRP) Ab (Stratech (Jackson ImmunoResearch), 323-165-021-JIR) at 1:100. PBTX (PBS + 0.1% Triton X-100 + 0.1% BSA) was used throughout for rinsing and incubating. Neuromuscular junctions at larval ventral longitudinal muscles 6/7 were imaged using the Olympus FV1000 confocal microscope (Tokyo, Japan). 

All quantitative image analyses were performed using ImageJ (National Institutes of Health, Bethesda, MD, USA). GluRIIA abundance was measured in maximum intensity projection images by quantifying the mean postsynaptic immunofluorescence intensity in relation to the fluorescence in the surrounding muscle tissue (Fsynapse-Fbackground) [37]. NMJ area, NMJ length, 1b bouton area and 1b bouton number were normalised to the area of muscles 6/7. 

### 2.5. Electrophysiology

Two-electrode voltage clamp (TEVC) recordings were performed as described previously [38,39]. Sharp-electrode recordings were made from ventral longitudinal muscle 6 (m6) in abdominal segments 2 and 3 of third-instar larvae using pClamp 10.5, an Axoclamp 900A amplifier, and Digidata 1550B (Molecular Devices, San Jose, CA, USA) in haemolymph-like solution 3 (HL-3) [40]. Recording electrodes (50–70 MΩ) were filled with 3 M KCl. All synaptic responses were recorded from muscles with input resistances ≥4 MΩ, holding currents <4 nA at −60 mV and resting potentials more negative than −60 mV at 25 °C. Holding potentials were −60 mV. The extracellular HL-3 contained (in mM): 70 NaCl, 5 KCl, 20 MgCl_2_, 10 NaHCO_3_, 115 sucrose, 5 trehalose-hydrate, 5 HEPES, and 1.5 CaCl_2_. Average single evoked excitatory junction potential (eEJC) amplitudes (stimulus: 0.1–0.5 ms, 1–5 V) were based on the mean peak eEJC amplitude in response to 10 presynaptic stimuli (recorded at 0.2 Hz). Nerve stimulation was performed with an isolated stimulator (DS2A, Digitimer (Hertfordshire, UK)). All data were digitised at 10 kHz and for miniature event recordings, 120 s recordings were analysed to obtain mean miniature EJC (mEJC) amplitudes and frequencies. Both mEJC and eEJC recordings were off-line low-pass filtered at 500 Hz and 1 kHz, respectively. 

### 2.6. Statistical Analyses

Statistical analyses of behavioural phenotype, NMJ morphology and electrophysiology data were performed using GraphPad Prism 9 software. One-way ANOVA tests were used for comparisons between two or more groups, followed by Bonferroni post hoc tests. Mann–Whitney U tests were used on non-parametric data for comparisons between two groups. In all cases, *p < 0.05* is considered statistically significant (* *p* < 0.05, ** *p* < 0.01, *** *p* < 0.001, **** *p* < 0.0001, n.s. = not significant). Values are reported as the mean, with error bars indicating SEM. 

### 2.7. Metabolomics

#### 2.7.1. Solvents and Reagents

The used solvents and chemicals, methanol, water, acetonitrile, ammonium formate and methyl-tertiary butyl ether (MTBE), were all LC-MS grade and obtained from Fisher Scientific or Sigma-Aldrich. Tripentadecanoin and L-valine 13C515N (95%) were used as internal standards (for organic and aqueous phases, respectively); both were purchased from Sigma-Aldrich (Burlington, MA, USA).

#### 2.7.2. Metabolite Extraction

Third-instar wandering larvae were manually dissected in ice-cooled phosphate buffered saline on a dissection plate. Larvae were filleted and the internal organs (gut and central nervous system) removed and collected for metabolomic analyses. The anterior and posterior of the larvae were removed, and the body wall muscles were collected for muscle samples. Extraction of metabolites from each tissue was performed using a modified version of a method published previously [41,42]. Briefly, a ball bearing (4 mm steel) was added to the tube containing the sample and 30 µL of methanol. Samples were subsequently vortexed for 5 min to mechanically homogenise. To the homogenate, 5 µL of aqueous phase internal standard solution (2.5 mM L-valine 13C515N in 80:20 MeOH:H_2_O), 140 µL of MTBE containing 15 µM of tripentadecanoin (organic phase internal standard) and 10 µL of methanol were added to all samples, in this order. All samples were then incubated at 10 min at 4 °C to allow for cell membranes to be disrupted. Subsequently, samples were transferred to a HPLC vial with a 250 µL glass insert to which 40 µL of 0.15 mM ammonium formate in water was added, samples were then centrifuged at 5000× *g* for 5 min. A pooled quality control sample was created by pooling 5 µL of all analytical samples, with extraction blanks produced by replacing larval tissue with 15 µL of HPLC grade water and applying the same extraction protocol. 

#### 2.7.3. Hydrophilic Liquid Interaction Chromatography (HILIC) Analysis of Aqueous Phase

Chromatography was performed using Agilent infinity HPLC system, with aqueous phase metabolites separated using 10 mM ammonium formate in water as mobile phase A and 2.5 mM ammonium formate in acetonitrile as mobile phase B on an Agilent Poroshell HILIC-z column (2.1 × 150 mm, 4 μM). The column was maintained at 30 °C and the solvent flow was 0.25 mL/min; for the first 5 min, the gradient was held isocratic at 95% mobile phase B, followed by a linear increase to 90% at 6 min, and 75% by 15 min. Subsequently, the column was cleaned for 3 min at 20% mobile phase B prior to restoration of initial conditions to allow the column to re-equilibrate for 7 min. Mass spectrometry was performed on an Agilent 6550 ion funnel QToF (Agilent, Santa Clara, CA, USA) with data collected between 50 and 1000 *m*/*z*, the drying gas flow was set to 15 L/min, a nebulizer pressure of 40 psi, a gas temperature of 200 °C, the sheath gas temperature was 300 °C and a flow of 12 L/min. 

#### 2.7.4. Reversed Phase Analysis of Non-Aqueous Phase

Organic phase metabolites were separated using 10 mM ammonium formate in water as mobile phase A and 10 mM ammonium formate in methanol:MTBE (2:1 *v*:*v*) as mobile phase B on an Agilent Poroshell C18 column (2.1 × 150 mm, 2.7 μM). The column was maintained at 55 °C and the solvent flow was 0.50 mL/min; initial gradient conditions were 80% mobile phase B. This was followed by a linear increase to 93% by 13 min, 94% by 20 min and 96% by 24 min prior to the column being cleaned for 6 min using 100% mobile phase B. Following restoration of initial conditions, the column was allowed to re-equilibrate for 5 min before the next injection. Mass spectrometry was performed on an Agilent 6550 ion funnel QToF (Agilent, Santa Clara, CA, USA) with data collected between 50 and 1200 *m*/*z*, the drying gas flow was set to 15 L/min, a nebulizer pressure of 35 psi, a gas temperature of 200 °C, the sheath gas temperature was 120 °C and a flow of 10 L/min.

#### 2.7.5. Data Processing and Statistical Analysis

Proteowizard [42] was used to convert the generated .d files into the .mzXML format for processing. These .mzXML files were processed in R (v3.6.0), using the CAMERA package; the “centwave” method was used for peak identification and integration as this allows the deconvolution of slightly overlapping and closely eluting peaks [43,44]. Once the peak output table had been generated, the data were filtered; for a measured feature to be retained in the dataset for further analysis, the peak had to be present in all samples of at least one group and have an abundance at least 5 times higher in analytical samples relative to extraction blanks.

A range of multivariate statistical approaches were applied to the combined dataset (HILIC and RP data) with all analysis performed in SIMCA (v13.0.4), and prior to analysis, data was scaled to unit variance (UV) and logarithmically transformed (base 10). Principal component analysis (PCA) was applied to the data to identify potential outliers and to explore macro trends within the dataset, with subsequent descriptive and predictive modelling of the class variable performed using partial least squares discriminant analysis (PLS-DA). Model performance was assessed based on the cumulative correlation coefficients (R2X(cum)) and predictive performance based on 7-fold cross validation (Q2(cum)), with the significance of the model assessed based on the ANOVA of the cross-validated residuals (CV-ANOVA).

Prior to univariate analysis, the normality of our data was determined using a Shapiro–Wilks test with normally distributed data analysed using a t-test and skewed data with a Mann–Whitney test; all analyses were performed in R (v3.6.0). Initial metabolite annotation was performed using an in-house library of metabolite standards, based on matching of accurate mass, retention time and fragmentation spectra. If no match was found publicly available, spectral libraries including HMDB (hmdb.ca) and METLIN (metlin.scripps.edu) were searched to identify putative annotations. 

## 3. Results

### 3.1. NA and 4-MOA Partially Reverse the Impaired Motor Phenotype in C9-Model Larvae 

To assess the possible effects of MCFAs on motor function, we quantified crawling speed and body contraction frequency in third-instar C9 larvae. We overexpressed 100 copies of the arginine-rich (GR) dipeptide repeat protein (DPR) arising from the hexanucleotide repeat expansion in *C9orf72* (*UAS-GR100*), using a motor-neuronal driver (*D42-GAL4*) to create C9-model larvae (*D42/GR100*). Previously, larvae expressing GR100 in the motor neurones using *OK6-GAL4* exhibited a 70–80% reduction in larval crawling speed compared to controls [31,35]. Although both are motor neuronal drivers, *OK6-GAL4* expression is more restricted to motor neurones compared to *D42-GAL4,* which also drives low levels of expression in the peripheral nervous system, including body wall sensory neurons [45]. Despite driver differences, we found that motor function was significantly diminished in *D42/GR100* larvae compared to *D42-GAL4/+* control animals (Figure 1A–D). C9 larvae displayed ~84% reduction in crawling speed and ~85% reduction in body wall contraction frequency, compared to control larvae (Figure 1A–D). 

We evaluated MCFAs for their ability to improve motor function in *D42/GR100* larvae. As little is known about optimal dosing of these compounds, a range of concentrations between 50 μM and 5 mM of 4-MOA and NA were added to standard fly food and fed to C9 larvae. Food containing equal volumes of DMSO was used as a control, to ensure DMSO had no effect on the larvae. We observed that both fatty acids partially rescued crawling speed and body contraction frequency in *D42/GR100* larvae, at 100 μM 4-MOA and 5 mM NA, respectively (Figure 1A–D). The dosage of 100 μM 4-MOA increased crawling speed in *D42/GR100* larvae by ~55% and body contraction frequency by ~57%, but also increased the performance of the control larvae by 20–30%, suggesting that 4-MOA may not have a disease-specific effect (Figure 1A,B). On the other hand, 5 mM NA improved crawling speed in *D42/GR100* larvae by ~63% and body contraction frequency by ~68%, whilst having no significant effect on control animals, indicating an effect specific to our ALS model (Figure 1C,D). Overall, we concluded that both 4-MOA and NA significantly increase motor performance in our ALS model. 

### 3.2. 4-MOA Rescues Presynaptic Neurotransmitter Release in C9-Model Larvae 

Having established that 4-MOA and NA treatment improved the impaired motor phenotype in our C9 model, we went on to investigate the electrophysiological effects of these compounds at the neuromuscular synapse. In Drosophila, there are two presynaptic vesicle release modes that represent two different functional outputs at the NMJ: miniature excitatory junctional currents (mEJCs), and evoked excitatory junctional currents (eEJCs) [46]. Presynaptic, action-potential (AP)-evoked, Ca^2+^-dependent vesicular release is reflected by eEJC amplitude, whereas the mEJC frequency largely correlates with the number of synapses [47], and mEJC amplitude represents predominantly postsynaptic sensitivity to glutamate, largely determined by the proportion of different subunits of glutamate receptors (GluRs) [48]. Previous work showed that GR100 motor neuronal expression (using the *OK6-GAL4* driver) leads to a significant decrease in the amplitude of evoked responses and frequency of mEJCs, indicating a reduced presynaptic Ca^2+^-dependent vesicular release [35]. 

When NMJs were stimulated at a rate of 1 Hz to generate eEJCs, we observed a significant reduction in amplitudes in C9 larvae compared to *D42/+* (Figure 2B). Treatment of C9 larvae with 5 mM NA or 100 μM 4-MOA resulted in larger eEJC amplitudes statistically indistinguishable from *D42/+*, suggesting that both treatments rescue functional deficits at the synapse (Figure 2B). However, only the 4-MOA treatment resulted in a significantly higher response compared to the one measured in non-treated GR100-expressing larvae, indicating a full rescue of this phenotype (Figure 2B). When quantifying spontaneous (mEJC) activities, we revealed no change in mEJC amplitudes or spontaneous firing frequencies in larvae expressing GR100 in the motor neurones, compared to the control (Figure 2C,D). Although NA treatment significantly increased mEJC frequency compared to untreated C9 larvae, both NA and 4-MOA have no overall effect on spontaneous activities at the NMJ in C9 larvae (Figure 2C,D). Our data demonstrate that 4-MOA treatment rescues defective presynaptic AP-evoked vesicular release in the C9 model; however, neither compound has an effect on postsynaptic GluR functionality. Our results suggest that, although NA appears to have some effect on evoked responses, only 4-MOA was able to fully rescue physiological NMJ defects in this ALS model. 

### 3.3. 4-MOA and NA Modify Neuromuscular Junction Morphology in C9-Model Larvae 

Our electrophysiological and behavioural data prompted us to perform detailed analyses of pre- and postsynaptic structures at the C9 larval NMJ, to determine the effects of 4-MOA and NA on NMJ morphology. The *Drosophila* larval musculature consists of bilaterally symmetrical hemisegments in a repeated pattern along the body wall, each consisting of 30 unique postsynaptic longitudinal and oblique muscle cells [49]. We centred our analyses on easily accessible ventral longitudinal muscles 6 and 7, which are innervated by two axons to form a singular glutamatergic NMJ [50] (Figure 3A). 

We first quantified the density of GluRs on the postsynaptic side of the excitatory NMJ. GluRs at the *Drosophila-*larval NMJ have two distinct subtypes: type IIA (GluRIIA) which generates larger excitatory synaptic currents and mediates long-term plasticity in flies [51], and type IIB (GluRIIB) which tends to be less responsive to vesicular neurotransmitter release [48]. We focused our analyses on GluRIIA receptors, the main determinant of postsynaptic responses at this synapse [48], by quantifying GluRIIA signal intensity. We demonstrated a two-fold increase in GluRIIA intensity in *D42/GR100* larvae (Figure 3), consistent with previous findings [35]. The increased GluRIIA density may be compensatory for the diminished presynaptic release seen at this synapse. Taken together with our finding that postsynaptic glutamate sensitivity is unchanged with GR100 expression, our results indicate the presence of an elevated amount of non-functional GluRIIAs at the NMJ in C9 larvae. Treatment with NA significantly decreased the GluRIIA signal intensity in *D42/GR100* larvae, returning the intensity to that of the control larvae (Figure 3). Treatment of *D42/GR100* larvae with 4-MOA reduced GluRIIA intensity by ~50%, although this effect was not statistically significant (Figure 3). These results agree with our finding that NA has a disease-specific effect on motor dysfunction in C9 larvae, suggesting that NA more successfully improves both locomotor and synaptic dysfunction in the ALS model. 

We then quantified NMJ morphology in C9-model larvae. *Drosophila* models of C9ALS/FTD exhibit neurodegeneration at the NMJ, characterised by a decrease in type Ib (large) synaptic bouton number and area, total NMJ area and total 6/7 muscle area [31,35]. However, overexpression of GR100 in *Drosophila* motor neurones results in a significant reduction in synaptic bouton number without a change in muscle area [35]. We confirmed that GR100 expression in *Drosophila* motor neurones decreases the total NMJ area, and demonstrated that the NMJ length is also decreased compared to controls (Appendix A). We found no significant difference in the number of branches at the C9-larval NMJ compared to control animals (Appendix A). Our results also confirm that overexpression of GR100 in the motor neurones results in a significant reduction in type Ib bouton number (Figure 4A,B); however, we saw no significant differences in total bouton area per NMJ (Figure 4C,D). Our data suggest that increased bouton area offsets the reduction in bouton number, indicating the absence of compensatory mechanisms at this level. We also saw an increase in total 6/7 muscle area with GR100 expression (Figure 4E). It is possible that increased muscle size is compensatory for the reduction in presynaptic release seen with GR100 expression. The differences between our results and the results of previous work may also be due to differences between motor-neuronal drivers, as past studies used *OK6-GAL4* [35]. 

Treatment of *D42/GR100* larvae with 100 μM 4-MOA and 5 mM NA modified the morphological defects at the NMJ. Both compounds decreased muscle 6/7 area and increased the number of Ib synaptic boutons in the C9 model to match the control animals, but only NA rescued the synaptic bouton area (Figure 4B,C,E). Neither compound had an effect on the NMJ area or length (Appendix A). These results indicate that the number and area of synaptic boutons may play a larger role in determining muscle function in *Drosophila* C9 models than total NMJ area and length. 

### 3.4. 4-MOA and NA Alter Metabolism in C9-Model Larvae 

As the synaptic phenotype does not fully explain the rescue in motor function, we decided to analyse the metabolomic profile in C9-model larvae treated with NA and 4-MOA, to determine whether these compounds affect their metabolic status. In this study, a total of 8744 metabolite features were detected with 4432 measures from the organic fraction and 4312 from the aqueous phase. Principal component analysis (PCA) was used to check data quality and identify potential outliers. As a result of this analysis, one observation (central nervous system samples taken from an N-treated larvae) was removed as an outlier. Macro metabolic trends were explored using partial least squares discriminant analysis (PLS-DA) with genotype and both 4-MOA and NA treatments having a significant impact on metabolism in whole larvae, CNS (central nervous system), gut and muscle tissues (Figure 5). When looking at the results in the CNS (R2X = 0.474 R2Y = 0.973 Q2 = 0.366 CV-ANOVA = 0.0001), gut (R2X = 0.469 R2Y = 0.963 Q2 = 0.531 CV-ANOVA = 1.59 × 10^−6^) and muscle (R2X = 0.548 R2Y = 0.875 Q2 = 0.753 CV-ANOVA = 3.02 × 10^−9^) similar metabolic patterns are observed. A clear separation between the control (*D42/+*) and *D42/GR100* groups is seen on t(1) with both 4-MOA and NA significantly affecting metabolism, creating a distinct treated metabotype without rescuing pathology associated effects, as on t [1] they are no closer to *D42/+* than the untreated *D42/GR100*. However, when we look at the results from the whole larvae (R2X = 0.547 R2Y = 0.944 Q2 = 0.819 CV-ANOVA = 6.47 × 10^−9^), both MCFAs appear to be partially rescuing disease-associated metabolic dysregulations, as shown by their intermediate position between *D42/+* and *D42/GR100* on t [1] (Figure 5). 

Having shown that both 4-MOA and NA are partially rescuing ALS-associated metabolic dysregulation in whole larvae, we identified a metabolic circuit linking the GABA glutamate shunt, nicotinamide/nicotinate metabolism, aspartate, alanine and asparagine metabolism, the urea cycle and pyrimidine metabolism as driving these shifts (Figure 6). Of the 15 metabolites measured in this circuit, 12 were significantly (*p* < 0.05) different between *D42/+* and *D42/GR100* (Figure 7 and Appendix A) with only uracil, cytosine and ornithine unaffected. In total, the abundance of 11 of these metabolites was modified by both 4-MOA and NA with 10 of the 11 normalising the metabolic shifts seen in untreated larvae (Figure 7 and Appendix A). Of the remaining four metabolites, three responded to NA treatment whilst one responded specifically to 4-MOA treatment (Appendix A). 

Having identified a dysregulated metabolic circuit in the whole larvae, we looked at these pathways in specific tissues and found that the ALS pathology (*D42/GR100*) was affecting the metabolic circuit in the CNS and muscle with the abundance of five and nine metabolites altered, respectively (Appendix A), although the shifts were not always in the same direction as seen in whole larvae. These metabolic processes appeared to be unaffected in the gut with no metabolites being significantly associated with the ALS pathology (Appendix A). MCFA treatment had less impact on these metabolic pathways in the muscle and CNS, compared to on the whole larvae. In the CNS, 4-MOA rescued nicotinate and glutamate abundances and NA exacerbated nicotinate levels (Appendix A), whereas 4-MOA rescued the abundance of cytosine, nicotinate and uracil and exacerbated cytidine and NA rescued aspartate and exacerbated ornithine levels in the muscle (Appendix A). Our results demonstrate that GR100 expression has a significant effect on metabolism in *Drosophila* larvae. We have also shown that MCFAs appear to rescue this metabolic dysregulation in whole larvae; however, their effects are less pronounced in specific tissues. 

## 4. Discussion

The therapeutic potential of MCFAs has been demonstrated for several neurodegenerative diseases, such as Alzheimer’s disease [20] and Parkinson’s disease [21]; however, there is currently limited clinical evidence supporting MCFA-based treatments for ALS. To understand whether MCFAs affect ALS phenotypes, we performed detailed analyses of motor function, NMJ morphology and NMJ electrophysiology in *Drosophila* models of C9ALS/FTD. Overall, we demonstrated that two MCFAs, 4-MOA and NA, could be beneficial in combatting two of the major ALS symptoms: motor impairment and synaptic dysfunction. Both MCFAs partially restored the diminished motor function and reversed NMJ degeneration in C9 larvae. However, these two compounds do not have identical mechanisms of action; NA impacts NMJ morphology whilst 4-MOA affects NMJ physiology. Our results led us to analyse the metabolic profile of C9 larvae to determine the effects of presynaptic GR100 expression on metabolic networks, and the consequences of MCFA treatment. We showed that GR100 expression has a significant impact on metabolism in *Drosophila* larvae, and that both 4-MOA and NA partially correct this ALS-associated metabolic dysregulation. Together, our results identify 4-MOA and NA as modulators of motor function, NMJ morphology and physiology, and metabolism in ALS model flies. 

Although the MCT-KD is well characterised in the literature, relatively little is known about the mechanisms of action of 4-MOA and NA, especially in the context of neurodegenerative disease. These MCFAs were initially identified as potential epilepsy treatments, as they were able to reproduce the effects of valproate, a commonly prescribed epilepsy drug, on phosphoinositide signalling [24], and demonstrated anti-seizure activity in an epilepsy model [25]. 4-MOA is an MCFA with an 8-carbon backbone, and is a derivative of OA, differing only by an additional methyl group on the fourth carbon atom. 4-MOA has previously been shown to be a more potent treatment for drug-resistant epilepsy than the current treatment, OA [52]. 4-MOA also provided neuroprotection in an excitotoxic cell death model, but OA was unable to do the same, again showing that 4-MOA has increased efficacy compared to OA [52]. Some forms of epilepsy can be neurodegenerative, suggesting that 4-MOA may provide neuroprotection in other neurodegenerative diseases, such as ALS. As 4-MOA is more potent than OA as a treatment for epileptic seizures, it is reasonable to hypothesise that they are functionally related, and that 4-MOA may be more efficient in other cases where OA is used. On the other hand, NA is an MCFA with a 9-carbon backbone, and is far less characterised in the literature. NA has antifungal [53] and antibacterial [54] properties, and is commonly used in cosmetics [55]; however, no data so far have demonstrated its potential as a treatment option for human neurodegenerative disease. NA is known to stimulate the differentiation of neuroblastic cell lines into neurones by inducing neurite growth; OA was also able to stimulate neuroblast differentiation, indicating a shared cellular mechanism [56]. As both 4-MOA and NA share respective functions with OA, it is possible that 4-MOA and NA are also functionally related. 

Our behavioural data implicate a shared function for 4-MOA and NA, as both MCFAs partially restored impaired crawling ability and body contraction frequency in C9 larvae. However, as 4-MOA also increased motor performance in control animals, NA appears to have a more specific effect on ALS pathologies. Similar results have been observed in a SOD1 mouse model of ALS, in which OA treatment improved locomotion [23]. The positive effect of MCFA treatment in two different model systems suggests that MCFA treatment may be beneficial for different types of ALS. 

NMJ defects are prominent in ALS patients, and NMJ dysfunction likely precedes motor neurone loss [14,15,16,17]; the ability to modify these symptoms may therefore be key in both the preventative and therapeutic treatment of ALS. Morphological and electrophysiological defects at the NMJ in C9 larvae have been well characterised [31,35], and our results agree with the majority of previously published work. However, we demonstrated an increase in larval muscle area and synaptic bouton area following motoneuronal GR100 expression, contrary to the previous work that reported a decrease in these parameters [31,35], possibly due to a sightly broader expression profile of *D42-GAL4* compared to the *OK6-GAL4* line used in the aforementioned study [45]. We also expected to see a decrease in both evoked amplitude and the frequency of miniature responses at the NMJ with motor neuronal GR100 expression, reflecting defects in Ca^2+^-dependent, AP-evoked vesicular release [35]. Although we confirmed a decrease in evoked amplitude, we did not detect any change in spontaneous firing frequencies in C9 larvae, indicating that GR100 expression does not affect spontaneous vesicular release. MCFA treatment impacts both NMJ morphology and physiology in C9 larvae, with each compound having a slightly different effect. Both 4-MOA and NA rescued the muscle 6/7 area and synaptic bouton number in C9 larvae, whereas only NA fully rescued the bouton area and postsynaptic GluRIIA density. When analysing synaptic function, we found that both compounds had an effect, but only 4-MOA was able to fully rescue electrophysiological deficits. Overall, our results suggest that NA has a stronger effect on NMJ morphology in C9 larvae, whereas 4-MOA predominantly affects NMJ functionality. 

Once we demonstrated the ability of NA and 4-MOA to rescue ALS phenotypes in our C9 model, we wanted to elucidate the mechanism of action of these compounds using metabolomic analyses. We revealed that presynaptic GR100 expression has a significant impact on metabolism in whole larvae, and in the CNS, gut and muscle tissue. MCFA treatment partially rescues ALS-associated metabolic dysregulation, acting on multiple linked metabolic pathways, and normalising 10 of the 12 metabolites measured as significantly different with GR100 expression. 

Our results indicate that GR100 expression results in highly elevated glutamate levels in whole larvae, possibly leading to excitotoxicity, a key ALS pathology. Glutamate receptors can become overstimulated by high levels of extracellular glutamate, resulting in an influx of Ca^2+^ ions that initiates neuronal cell death [57]. Dysregulation of glutamate cycling genes is reported in astrocytes isolated from ALS patients and accompanied by increased intracellular glutamate level [58]. We showed that MCFA treatment normalised glutamate levels in C9 larvae; alongside our findings that these compounds rescue increased postsynaptic GluRIIA density, our data indicate that 4-MOA and NA may rescue glutamate excitotoxicity. These findings agree with the observation that acetone and β-hydroxyburate (BHB), two of the main ketone bodies produced via MCFA hydrolysis, directly inhibit glutamatergic NMDA receptors in frog oocytes [59]. Acetoacetate, the third ketone body produced by MCFA metabolism, has also been demonstrated to reduce glutamatergic neurotransmission at the *Drosophila* NMJ by decreasing presynaptic glutamate release [60]. 

We also noted that GR100 expression dysregulated the urea cycle, seen via a significant increase in arginine and citrulline, and a significant decrease in ornithine and urea in untreated C9 larvae. The urea cycle functions to eliminate excess ammonia and nitrogen; a dysfunction in this process causes increased ammonia production in skeletal muscle, resulting in chronic hyperammonaemia [61]. Ammonia is able to cross the blood–brain barrier (BBB) in its gaseous form, where it can induce neuroinflammation and increase glutamatergic transmission [62]. Hyperammonaemia is also known to induce mitochondrial dysfunction in animal models, resulting in neuronal cell death [63]. Importantly, ammonia neurotoxicity has been proposed as a key feature in ALS pathogenicity [64]. Evidence of hyperammonaemia in ALS has been provided by a clinical study, noting that levels of ammonia correlate with disease duration [65]; this link was further confirmed in an SOD1 mouse model of ALS [66]. We hypothesise that hyperammonaemia may be contributing to the ALS pathology in our C9 model. Importantly, 4-MOA and NA treatment returned levels of urea cycle metabolites to control levels, suggesting the capacity of these compounds to combat motor and NMJ dysfunctions by rescuing ammonia levels. In line with this hypothesis, sodium phenylbutyrate, a drug used to treat increased ammonia levels in patients with urea cycle disorders [67], has recently been approved for use in ALS patients [68], after demonstrating its ability to promote motor neuronal survival in an SOD1-ALS-mouse model [69]. Although this drug is thought to be beneficial in ALS by reducing toxicity from endoplasmic reticulum stress [70], it is possible that the reduction of ammonia may also be clinically relevant. 

L-arginine (L-Arg) is a metabolically active amino acid that can be metabolised via nitric oxide synthase (NOS), into nitric oxide (NO) and L-citrulline. NO is a small, gaseous molecule with multiple roles in the CNS that include regulation of synaptic function, neuromodulation and neuromuscular junction formation [71,72,73]. NO facilitates neuronal death and neurodegeneration through glutamate-mediated excitotoxicity and inhibition of mitochondrial function [74,75,76]. In ALS, NO-mediated toxicity promotes degeneration of motor neurones [77]; this effect is likely mediated by the NO-derived peroxynitrite formation and subsequent 3-nitrotyrosination of proteins which is increased in the CNS of people affected by either sporadic or genetic forms of ALS [78], and in transgenic mouse models [79]. In line with the proposed role of NO in the ALS pathogenesis, pharmacological inhibition of NO release protects cultured motor neurones from cell death and glia-induced toxicity [80]. L-citrulline is a co-product in the generation of NO from L-Arg and a reliable marker of neuronal NO released sites [81]. Moreover, citrulline levels and transporter activity are dysregulated in experimental models of ALS [82]. Our data suggest a moderating effect of NA and 4-MOA on both arginine and citrulline—and therefore, likely, on NO levels—in whole larvae, with NA almost completely reversing the GR100-induced increase in arginine and citrulline to control levels. These data indicate another possible mechanism of action for these compounds in the context of ALS. 

Overall, we provide evidence that 4-MOA and NA can improve phenotypes in a *Drosophila* model of ALS. We illustrate that, although similar, these two MCFAs do not have identical mechanisms of action, with NA exerting more specific effects on ALS-related phenotypes compared to 4-MOA. We suggest these MCFAs may act on ALS pathologies by rescuing glutamate, arginine, citrulline and ammonia levels; further study is necessary to fully elucidate the workings of these compounds and their effects on ALS-related pathologies. 

## 5. Conclusions

A hexanucleotide repeat expansion in the *C9orf72* gene is the most common genetic cause of ALS/FTD. Medium-chain fatty acids—saturated fatty acids consisting of 6–12 carbon atoms—are an understudied class of molecules that can improve metabolic features as well as cognition in humans and as such hold promise for therapeutic interventions in ALS. Our work clearly demonstrated the ability of two medium-chain fatty acids—nonanoic acid (NA) and 4-methyloctanoic acid (4-MOA)—to reverse the impaired locomotion and synaptic function in a *C9orf72* Drosophila model of ALS. Our work also addressed one of the major problems in developing effective therapies for ALS—the lack of suitable targets. Our comprehensive metabolomic analyses of NA- and 4-MOA-treated flies revealed molecular targets for future interventions and provided novel insights into the physiology and pathogenesis of this disease.

## Figures and Tables

**Figure 1 cells-12-02163-f001:**
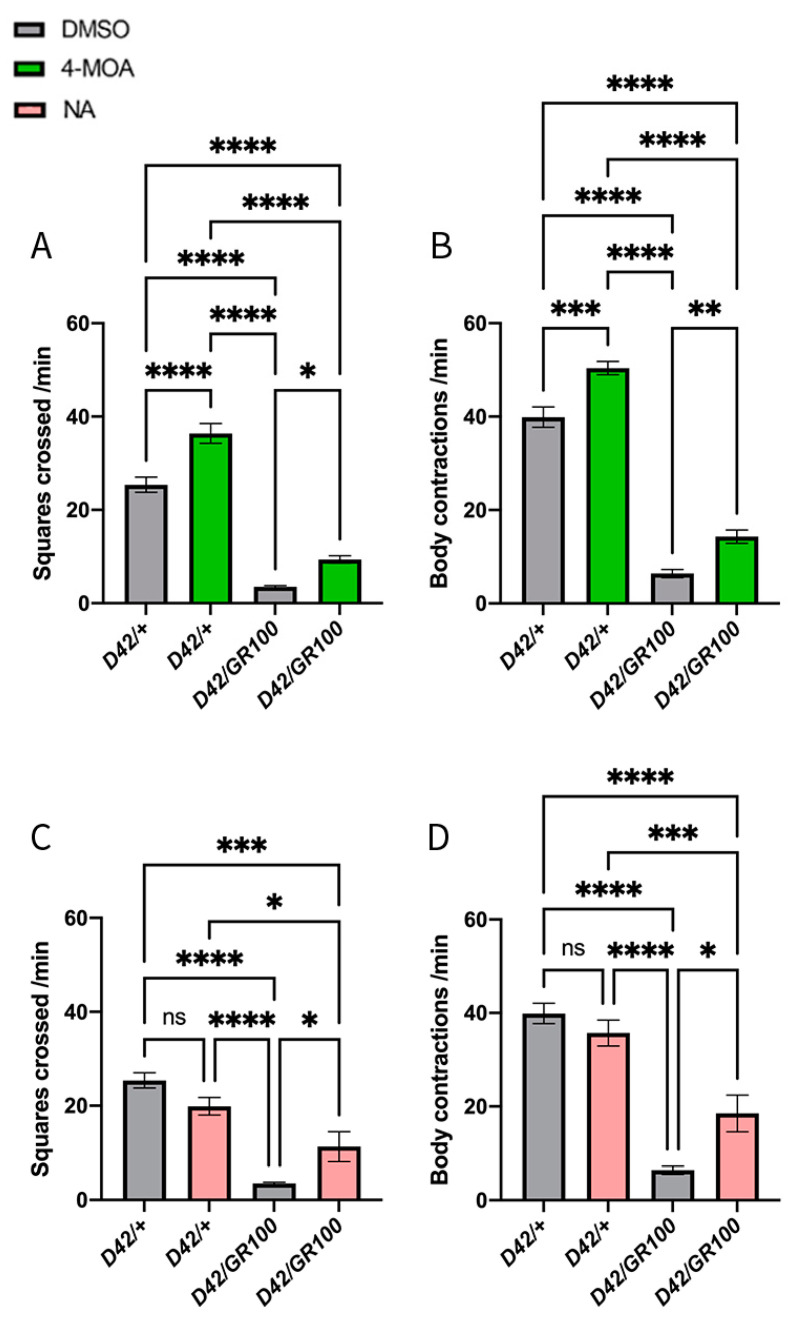
4-MOA and NA partially rescue behavioural phenotype in ALS flies. Crawling speed (**A**,**C**) and body contraction frequency (**B**,**D**) of ALS-model larvae (*D42/GR100*) expressing 100 copies of the ALS-associated arginine-rich (GR) DPR in the motor neurons, treated with 100 μM 4-MOA (green) and 5 mM NA (pink), compared to controls expressing only the motor-neuronal driver (*D42/+)* (*n* = 8–10). Control larvae were given equal volumes of DMSO (grey). *p <* 0.05 is considered statistically significant (* *p* < 0.05, ** *p* < 0.01, *** *p* < 0.001, **** *p* < 0.0001, n.s. = not significant).

**Figure 2 cells-12-02163-f002:**
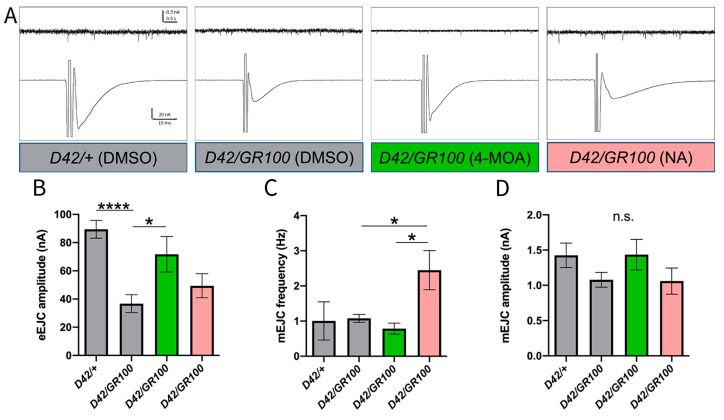
4-MOA treatment rescues degraded synaptic strength at the NMJ in ALS-model larvae. (**A**) Representative electrophysiological traces of evoked (eEJC) and spontaneous (mEJC) responses from motor-neuronal driver control (*D42/+*) and ALS (*D42/GR100*)-larval NMJs, and ALS-larval NMJs treated with 4-MOA (green) and NA (pink). Quantification of eEJC amplitude (**B**), mEJC frequency (**C**) and mEJC amplitude (**D**) for indicated genotypes and treatments (*n* = 5–9). Controls were given equal volumes of DMSO (grey). *p <* 0.05 is considered statistically significant (* *p* < 0.05, **** *p* < 0.0001, n.s. = not significant).

**Figure 3 cells-12-02163-f003:**
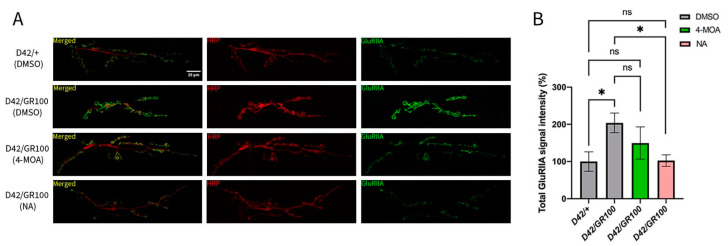
NA rescues augmented GluRIIA signal intensity in ALS-model larvae. (**A**) Representative confocal images showing the result of GR100 expression under *D42-GAL4* control at the NMJ (*D42/GR100)*, and the affect 4-MOA and NA treatment has on the NMJ. Anti-HRP (red) indicates motor neurones innervating muscles 6/7; anti-GluRIIA (green) shows the presence of postsynaptic GluRIIA receptors. (**B**) Total GluRIIA signal intensity at the third-instar larval NMJ in indicated genotypes, with 4-MOA (green) and NA (pink) treatment (*n* = 5–6). Controls were given equal volumes of DMSO (grey). *p <* 0.05 is considered statistically significant (* *p* < 0.05, n.s. = not significant).

**Figure 4 cells-12-02163-f004:**
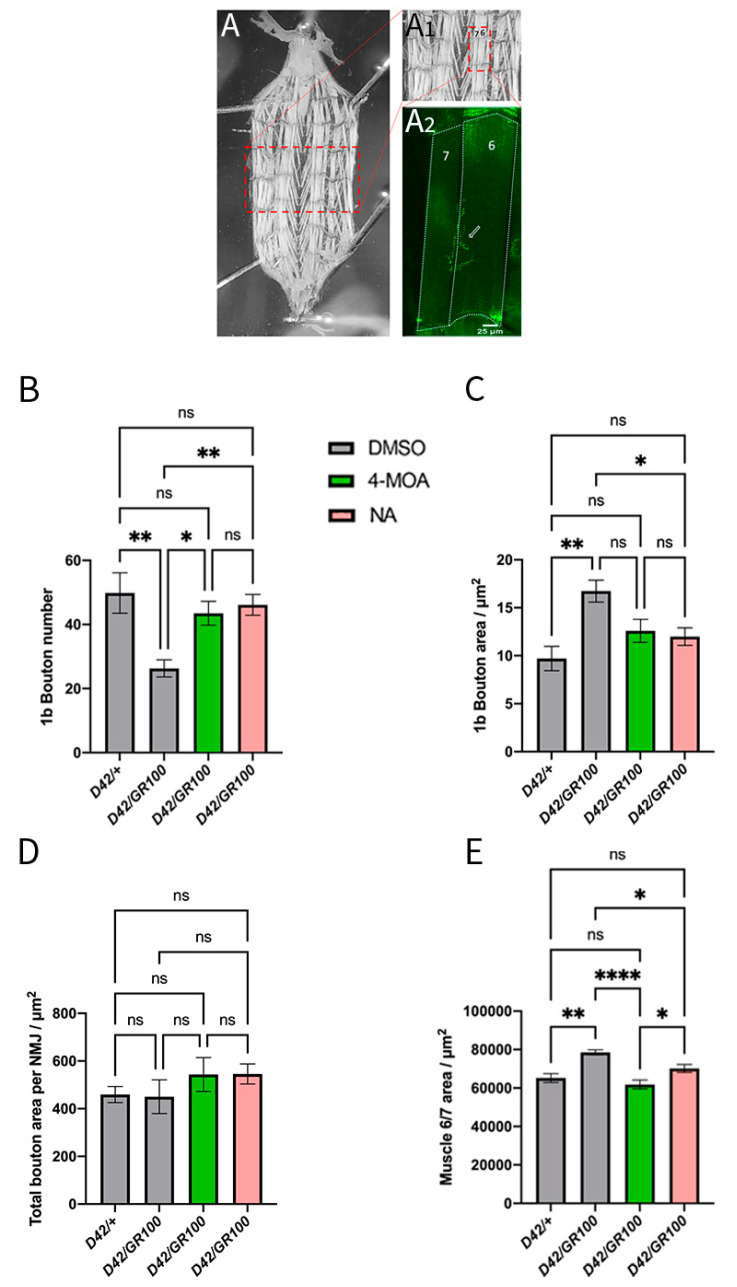
4-MOA and NA modify NMJ morphology in ALS-model larvae. (**A**) Images showing a dissected third-instar larvae under a dissecting microscope; red box (inset) shows the location from which NMJs were chosen to be imaged and analysed. (**A1**) Red box (inset) shows muscles 6/7 that are innervated by the NMJ. (**A2**) Representative confocal image of a single larval abdominal hemisegment containing muscles 6/7; arrow (inset) shows the location of the NMJ that innervates muscles 6/7. Area of muscles 6/7 was determined by measuring the respective area of muscles 6 and 7 (white outline) and calculating the average. Quantification of 1b synaptic bouton number (**B**), Ib (large) synaptic bouton area (**C**), total bouton area per NMJ (**D**) and muscle 6/7 area (**E**) in indicated genotypes, and with 4-MOA (green) and NA (pink) treatment (*n* = 5–6). Controls were given equal volumes of DMSO (grey). *p* < 0.05 is considered statistically significant (* *p* < 0.05, ** *p* < 0.01, **** *p* < 0.0001, n.s. = not significant).

**Figure 5 cells-12-02163-f005:**
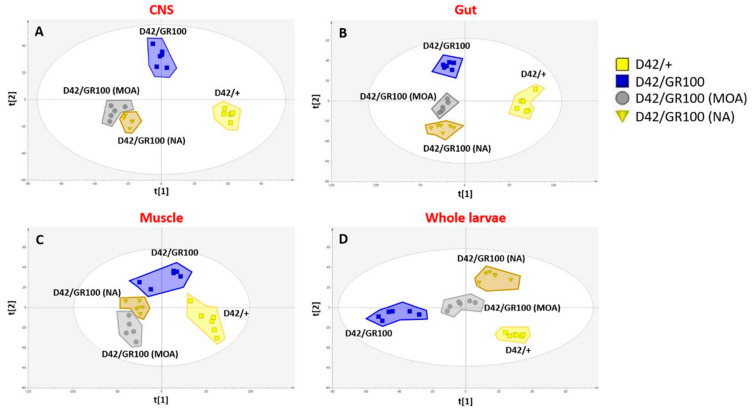
Multivariate partial least squares discriminant analysis (PLS-DA) exploring the effect of ALS pathology (*D42/GR100*) and 4-MOA and NA treatment. (**A**) PLS-DA score plot of all samples in CNS (R2X = 0.474 R2Y = 0.973 Q2 = 0.366 CV-ANOVA = 0.0001). (**B**) PLS-DA score plot of all samples in CNS gut (R2X = 0.469 R2Y = 0.963 Q2 = 0.531 CV-ANOVA = 1.59 × 10^−6^). (**C**) PLS-DA score plot of all samples in CNS muscle (R2X = 0.548 R2Y = 0.875 Q2 = 0.753 CV-ANOVA = 3.02 × 10^−9^). (**D**) PLS-DA score plot of all samples in whole larvae (R2X = 0.547 R2Y = 0.944 Q2 = 0.819 CV-ANOVA = 6.47 × 10^−9^).

**Figure 6 cells-12-02163-f006:**
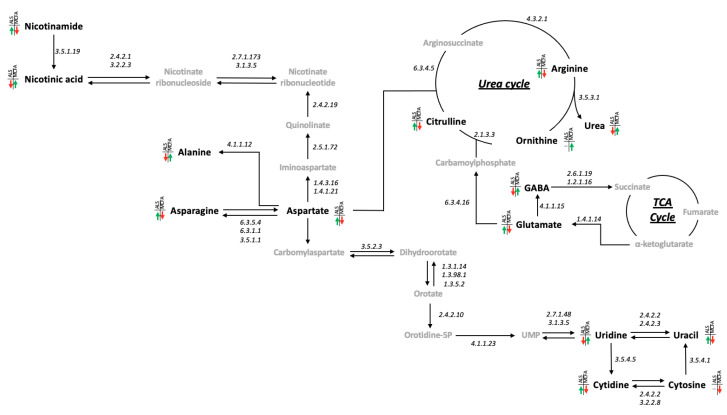
Metabolic circuit that is being dysregulated in ALS and rescued by MCFA treatment. Metabolites in “black” were measured in our data, metabolites in “grey” were not detected. Changes in abundance caused by ALS and MCFA treatment are shown; a red arrow denotes a reduced abundance, a green arrow denotes an increased abundance, and a grey line means no change was observed.

**Figure 7 cells-12-02163-f007:**
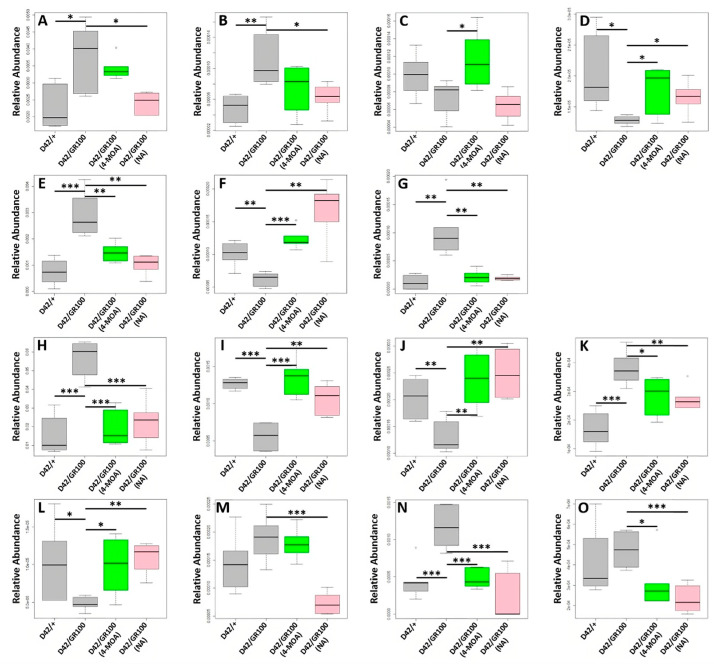
Boxplots showing the abundance of the 15 metabolites measured in the identified metabolic circuit. (**A**) Arginine, (**B**) citrulline, (**C**) ornithine, (**D**) urea, (**E**) glutamate, (**F**) GABA, (**G**) aspartate, (**H**) nicotinamide, (**I**) nicotinate, (**J**) alanine, (**K**) asparagine, (**L**) uridine, (**M**) uracil, (**N**) cytidine, (**O**) cytosine. * *p* < 0.05, ** *p* < 0.01, *** *p* < 0.001.

## Data Availability

Data are contained within the article and Appendix A. Additional data (e.g., metabolomics) are available from corresponding authors upon request.

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
