# Peer review of "Medium-Chain Fatty Acids Rescue Motor Function and Neuromuscular Junction Degeneration in a Drosophila Model of Amyotrophic Lateral Sclerosis"

_cells, 2023, doi:10.3390/cells12172163_

Round 1
Reviewer 1 Report
There is limited direct research available on the relationship between medium chain fatty acids and amyotrophic lateral sclerosis. Medium-chain fatty acids are a type of dietary fat that have gained attention for their potential health benefit. They are metabolized in different way compared to long-chain fatty acids, and some studies have suggested that they might have positive effects on metabolism and energy production.
This is an interesting study on ALS drosophila model that provides the evidence that 4-methyloctanoic acid and nonanoic can improve phenotypes in a Drosophila model of ALS. The relationship between MCFA and ALS remains unclear and needs further research on ALS murine models and patients.
Author Response
Nothing to respond to.
Reviewer 2 Report
To authors
I congratulate the authors for the quality of their work and for the importance of the results. I have suggestions and a question for the authors:
-Title: Do not add abbreviations. Describe NMJ.
-Abstract: DPRs. It does not describe before what it is. Describe the authors referring to dipeptide repeats, then add abbreviation.
- Introduction: line 36. "...measured using 6- or 10-minute walking tests [6,7]".
- According to the bibliography, do the authors refer to 6 minutes or 10 meters?
- Materials and Methods Section. Lines 116-117: "...expression via GAL4- dependent upstream activator sequence) (Brand and Perrimon, 1998)".
Authors must add this reference in the manuscript (reference section) and must cite it in Vancouver in the text.
- Lines 314-316: “Our data demonstrate that 4-MOA treatment rescues defective presynaptic Ca2+-dependent vesicular release in the C9 model, however, neither compound has an effect on post-synaptic GluR functionality”.
Figure 2 (A and B). It would have been interesting if the authors complemented the information with "Calcium Imaging" or other measurements of the calcium status of a cell. What do the authors think the release of neurotransmitters is dependent on calcium? There are also clear indications of channelopathies in ALS, including alterations in the expression of families of potassium channels dependent on pH, CO2 and/or voltage.
- Please, the authors should review reference 57, it appears incomplete.
Author Response
We would like to thank this reviewer for their insightful and comments and suggestions. Our point-by-point response is included below.
REVIEWER: I congratulate the authors for the quality of their work and for the importance of the results.
AUTHOR: We are very grateful for the reviewer’s positive and encouraging comments.
REVIEWER: Title: Do not add abbreviations. Describe NMJ.
AUTHOR: We modified the title by changing ‘NMJ’ to ‘neuromuscular junction’.
REVIEWER: Abstract: DPRs. It does not describe before what it is. Describe the authors referring to dipeptide repeats, then add abbreviation.
AUTHOR: We spelled out the acronym ‘DPR’ in the Abstract as suggested.
REVIEWER: Introduction: line 36. "...measured using 6- or 10-minute walking tests [6,7]". According to the bibliography, do the authors refer to 6 minutes or 10 meters?
AUTHOR: We are grateful to the reviewer for noticing this mistake. We corrected the sentence in line 36 (“6-minute and 10-meter walking test…”).
REVIEWER: Materials and Methods Section. Lines 116-117: "...expression via GAL4- dependent upstream activator sequence) (Brand and Perrimon, 1998)". Authors must add this reference in the manuscript (reference section) and must cite it in Vancouver in the text.
AUTHOR: We added the reference (Brand and Perrimon, 1998) and properly cited it in the main text.
REVIEWER: Lines 314-316 (now 321-323): “Our data demonstrate that 4-MOA treatment rescues defective presynaptic Ca2+-dependent vesicular release in the C9 model, however, neither compound has an effect on post-synaptic GluR functionality”.
Figure 2 (A and B). It would have been interesting if the authors complemented the information with "Calcium Imaging" or other measurements of the calcium status of a cell. What do the authors think the release of neurotransmitters is dependent on calcium? There are also clear indications of channelopathies in ALS, including alterations in the expression of families of potassium channels dependent on pH, CO2 and/or voltage.
AUTHOR: At virtually all known chemical synapses, including the Drosophila larval NMJ, presynaptic action potentials (APs) – the ‘evoking stimuli’ - activate voltage-gated calcium channels, allowing calcium to enter the motoneurone and trigger vesicle fusion and neurotransmitter release. For the purpose of clarity, i.e. to emphasise the stimulus-induced nature of these responses, we have replaced “Ca2+-dependent vesicular release” with “action-potential (AP)-evoked vesicular release” (line 322), or used both (lines 302 and 534). While calcium measurements are indeed very useful when studying mechanisms of presynaptic vesicle release, electrophysiology remains the gold standard for the direct, functional assessment of synaptic function, which is why we chose this approach.
The reviewer correctly noted that disruption of various ion channels closely associates with ALS pathogenesis, and it is possible that some ion channels are also functionally impaired in our ALS model. While further exploration of ion channel function in the context of ALS is undoubtedly relevant, it is worth noting that any disruption of e.g. potassium and sodium channels in the presynaptic (motoneuronal) compartment will almost inevitably lead to diminished calcium cycling and, consequently, reduced evoked response.
REVIEWER: Please, the authors should review reference 57, it appears incomplete.
AUTHOR: We modified/expanded reference #57 (now #58). We also corrected some other minor mistakes in the Reference section.